# A history-dependent integrase recorder of plant gene expression with single-cell resolution

Cassandra J. Maranas[1], Wesley George[1], Sarah K. Scallon[1,3], Sydney VanGilder[1,3], Jennifer L. Nemhauser [1] & Sarah Guiziou [2]✉

During development, most cells experience a progressive restriction of fate that ultimately results in a fully differentiated mature state. Understanding more about the gene expression patterns that underlie developmental programs can inform engineering efforts for new or optimized forms. Here, we present a four-state integrase-based recorder of gene expression history and demonstrate its use in tracking gene expression events in *Arabidopsis thaliana* in two developmental contexts: lateral root initiation and stomatal differentiation. The recorder uses two serine integrases to mediate sequential DNA recombination events, resulting in step-wise, history-dependent switching between expression of fluorescent reporters. By using promoters that express at different times along each of the two differentiation pathways to drive integrase expression, we tie fluorescent status to an ordered progression of gene expression along the developmental trajectory. In one snapshot of a mature tissue, our recorder is able to reveal past gene expression with single cell resolution. In this way, we are able to capture heterogeneity in stomatal development, confirming the existence of two alternate paths of differentiation.

The sequential expression of genes during development progressively restricts and specifies a cell's fate. Many of the genes involved in cell differentiation processes have been identified across a host of model and non-model organisms, and, in many cases, these genes have been assembled into cell fate pathways. However, a full understanding of development will require combining the establishment of these genetic regulatory networks with information about the extent and effect of variation in how these networks are experienced by individual cells. The key limiting technology to achieve this aim is a method to monitor gene expression history with the single-cell resolution that maintains spatial information.

Existing single-cell RNA sequencing (scRNAseq) technologies provide a wealth of gene expression data in a variety of contexts[1]. However, scRNAseq provides only a snapshot of the cell's transcriptional state at one point in time, fails to capture the expression of lowly expressed genes, and requires the destruction of the sample, thus losing spatial resolution. Recent methods combining scRNAseq with additional techniques (such as metabolically labeling mRNAs[2,3], evaluating mRNA splice status[4], and pseudo-time analysis[5]) have been developed to allow users to infer some temporal information and have been applied to study cell specification in *Zea mays*[6] and *Arabidopsis thaliana*[7]. As a different approach to gain more information from scRNAseq, efforts to add spatial information include positional barcoding in tandem with next-generation sequencing in mammalian[8] and plant[9] systems; and in situ sequencing combined with in situ fluorescent hybridization in mammals[10] and plants[11]. Such approaches provide some spatial context, but no temporal information and are quite costly and complex. Alternatively, keeping spatial and temporal information, methods that utilize fluorescent expression as a marker of genetic events have been widely used for decades in a diverse array

[1]Department of Biology, University of Washington, Seattle, WA, USA. [2]Engineering Biology, Earlham Institute, Norwich, UK. [3]These authors contributed equally: Sarah K. Scallon, Sydney VanGilder. ✉e-mail: sarah.guiziou@earlham.ac.uk

of organisms, including mouse models[12] and plants[13]. These approaches are limited in information due to the finite number of fluorescent tags that can be used at one time, the stress to the organism due to repeated irradiation, the long half-lives, the slow maturation time, and the photobleaching risk of fluorescent proteins. One previous study[14] implemented a fluorescence-based live imaging approach over multiple days for studying lateral root development in *Arabidopsis* but was able to track only six individuals due to the complexity of the imaging setup.

Recently, DNA-based recording systems have been developed and overcome some of the challenges of 'omic and microscopy-based technologies. Systems based on CRISPR-Cas9 mediated mutations and subsequent DNA sequencing enabled the recording of the Wnt signaling pathway in mammalian cells[15], although this approach lacked spatial resolution. DNA-based recording devices utilizing CRISPR have also been used for lineage tracing, for example, in reconstructing lineages in *Arabidopsis* tissues[16], zebrafish organs[17], and in the development of a lineage tracing mouse line[18]. Researchers have also utilized the DNA recombination abilities of integrases for similar DNA-based recording approaches. For example, the Cre-LoxP system has been employed to trace phloem and xylem vascular cell lineage in *Arabidopsis*[19]. Other work employs stochastic integrase recombination to generate barcodes which are then deconvoluted to construct cell lineages in mouse cell lines and *Drosophila melanogaster*[20]. Synthetic memory switches using tissue-specific integrase expression have recently been implemented in *Arabidopsis*[21,22]. More complex integrase circuits have been designed and implemented in bacterial cells[23–25], mammalian cells[26,27], and plant protoplasts[21,27].

Plants are particularly interesting organisms for studying the differentiation process as they continue to generate new organs throughout their lifetime. Decoding plant developmental programs can also inform the engineering of novel crops with enhanced properties to adapt to environmental stresses accelerated by climate change[28].

Here, we build on previous work using single serine integrase switches prototyped in *Nicotiana benthamiana*[29] and implemented in *Arabidopsis* for tracking the expression of genes underlying lateral root initiation[22]. Following the logic of a previously engineered recorder in bacterial cells[25], we engineer a four-state history-dependent integrase-based recorder of gene expression in *Arabidopsis*. We first validate the design of the recorder target using constitutive expression of two serine integrases: PhiC31 and Bxb1. We then test the efficacy of the circuit to record two well-known transcriptional trajectories, those occurring during the development of lateral roots and stomata. In both cases, we are able to capture the order of expression of two genes in individual cells. In addition, the recorder reveals heterogeneity in the gene expression history of mature stomata, highlighting the alternative differentiation pathways that they had experienced.

## Results

### An integrase-based history-dependent recorder functions in plants

Serine integrases mediate site-specific DNA inversions or excisions based on the presence and orientation of specific integrase sites (inversion when sites are in opposite orientation and excision when in the same orientation). Our integrase-based, history-dependent recorder of gene expression leverages these recombination abilities, with four possible fluorescent expression outputs, each indicative of a specific order of occurrence of signal inputs. The recorder is composed of two parts: a target (composed of integrase sites, fluorescent proteins, and gene regulatory elements) and an integrase construct (mediating integrase expression). We designed our target construct (Fig. 1a) for *Arabidopsis* with a strong constitutive p35S promoter[30] and three genes encoding fluorescent proteins: ER-localized mtagBFP2[31] (named BFP from here on), nuclear-localized mCherry[32] (named RFP

from here on), and nuclear-localized NeonGreen[33] (named GFP from here on). The promoter and reporter genes are flanked by integrase sites for the orthogonal serine integrases: PhiC31 (Fig. 1a, gray triangles) and Bxb1 (Fig. 1a, black triangles). The target has four possible DNA states which result in different expression outputs: State 0 (BFP expression), State 1 (RFP expression), State 2 (GFP expression), and State α (no fluorescent expression). The switches between states are order-dependent and mediated by the PhiC31 and Bxb1 integrases.

The 'PhiC31 then Bxb1' target allows tracking of the PhiC31 then Bxb1 lineage, switching progressively from State 0 to State 2 when PhiC31 is expressed before Bxb1. In this target, State 0 represents the initial DNA state with no integrase expression. Expression of PhiC31 integrase mediates the switch to State 1 via inversion of the BFP-RFP cassette, resulting in a switch from BFP to RFP expression. Subsequent expression of Bxb1 integrase in these State 1 cells mediates the subsequent switch from State 1 to 2 via inversion of the p35S promoter and results in GFP expression. If Bxb1 is expressed 'out of order' prior to PhiC31 in a cell, it excises 75% of the target, including the p35S promoter, corresponding to a switch from state 0 to state α and resulting in a loss of fluorescence (Fig. 1a). The integrase switches are irreversible and heritable, so descendent cells inherit the target DNA state from the mother cell.

To characterize this target, we used a constitutive promoter to drive each integrase separately, the promoter for PROTEIN PHOSPHATASE 2 A SUBUNIT 3 (pPP2AA3, commonly used as a control for qPCR[34]). pPP2AA3 is not as strongly expressed as other commonly used plant constitutive promoters such as pUBQ10 or p35S, making it a better match for the expression level of most developmental genes. We first transformed the target in *Arabidopsis*, to obtain a stable target line. We confirmed strong expression of only BFP in both root and leaf tissue (Fig. 1b, left; Fig. 1c, left) in the target plant line selected for the rest of the experiments. We subsequently transformed this plant line with a PhiC31 constitutively expressed construct. We observed a switch in expression to RFP in both root and leaf tissue (Fig. 1b, middle), as expected. To test the State 1 to State 2 switch in isolation, we designed a switched target construct where the starting state of the target is the DNA State 1. We confirmed that this switched target shows strong RFP expression in the leaf and root (Supplementary Fig.1). When transformed with a pPP2AA3-expressed Bxb1 construct, this switched target exhibited strong constitutive GFP expression in both the leaf and root (Fig. 1b, right) confirming the switch from State 1 to State 2. Transformation of the State 0 target with pPP2AA3-expressed Bxb1 resulted in a strong reduction in BFP expression in the T1 generation (Fig. 1c), as expected. To confirm the switch from State 0 to State α, we genotyped this seedling and confirmed the presence of the target in a DNA state corresponding to the length of the State α target (Supplementary Fig. 2).

In addition to the 'PhiC31 then Bxb1' target, we also generated a 'Bxb1 then PhiC31' target (Supplementary Fig. 3) wherein the Bxb1 then PhiC31 lineage is tracked, switching from State 0 to 1 then 2 when Bxb1 then PhiC31 are expressed sequentially. We validated the initial switches for this target, showing Bxb1 expression causes a switch to RFP expression, therefore from State 0 to State 1 in the target, and PhiC31 expression to no fluorescent expression, therefore, to State α of the target (Supplementary Fig. 3). We also developed and made publicly available an alternate target for tracking the PhiC31 then Bxb1 lineage in which the reporters are expressed with pUBQ10 instead of p35S. We designed and implemented history-dependent integrase targets in *Arabidopsis* with identifiable fluorescent outputs for the different DNA states.

### History-dependent recording of gene expression during lateral root initiation

For the next step in prototyping, we used well-characterized developmental promoters in the integrase construct to control the

**Fig. 1 | The history-dependent integrase circuit switches between fluorescent states based on the order of inputs. a** Schematic of the integrase target circuit. The target consists of two sets of integrase sites for the PhiC31 and Bxb1 integrases; three genes for fluorescent proteins (mtagBFP2 (BFP), mCherry (RFP), NeonGreen (GFP)); and a p35S strong constitutive promoter for fluorescent expression. The initial state of the target (State 0) results in the expression of mtagBFP2. Upon expression of PhiC31, an inversion causes the target to switch to State 1 and results in the expression of RFP. Subsequent Bxb1 expression causes another inversion, flipping the promoter direction and mediating the switch to State 2, resulting in GFP expression. Alternatively, the addition of Bxb1 in a cell with the State 0 target results in a DNA excision, causing a switch to State α and loss of fluorescence. **b** The PhiC31-Bxb1 integrase order mediates a series of switches from State 0 to 1 to 2.

(left) The State 0 target shows strong BFP expression in the leaf and the root. (middle) Constitutive PhiC31 expression mediates a switch of the target to State 1 and strong RFP fluorescence in leaf and root tissue. (right) Subsequent constitutive expression of Bxb1 mediates a switch from State 1 to 2 and results in GFP expression in the leaf and the root. **c** Expression of Bxb1 prior to PhiC31 mediates an 'out of order' switch to State α and loss of fluorescence. (left) The initial target in State 0 shows strong BFP expression in the leaf and the root. (right) Constitutive expression of Bxb1 with the target in State 0 results in a switch to State α and a loss in fluorescence in the leaf and root. The root and leaf tissue of $n = 10$ seedlings from at least 2 T1 lines were characterized for each of the target States 0, 1, 2, and α. Black scale bars in the brightfield images correspond to 100 μm (root) and 50 μm (leaf). Source data are provided as a Source Data file.

expression of the integrases. To note, the integrase recorder captures gene expression in a digital manner, where each gene is either expressed (1) or not expressed (0). This is in contrast to biological systems where gene expression is analog. Gene expression levels are continuous with a cell-type-specific distribution of expression around a mean. As a result, expression levels that might be considered background or basal can be sufficient for integrase activity. In practice, in the integrase recorder, the digital state 0 corresponds to low gene expression, and the digital state 1 corresponds to expression above a specific threshold. This threshold is dependent on the level of integrase activity, and tuning parts can be used to adjust the switching threshold for the expression level of the gene of interest[22]. The switch threshold is affected by factors such as integrase efficiency, promoter strength, and other factors affecting expression, such as the insertion location of the integrase construct. The approach we took for deciding which tuning parts to use starts with generating the integrase construct without any tuning parts. Then, based on the switch result with this construct, we added a tuning part if needed, either a nuclear localization signal tag[22] (NLS, shown to increase integrase switching efficiency) if underswitched or an RNA destabilization tag from SMALL AUXIN UP-REGULATED RNA genes[35] (DST) if overswitched.

For an initial test of the recorder, we turned to the development of lateral roots, as many promoters for genes involved in lateral root cell fate specification have been already characterized using reporter constructs[36,37] and, for a few promoters, using integrase switches[22]. Beyond the depth of knowledge about lateral root developmental programming, it is also of high interest for engineering efforts aimed at increasing drought resilience[38]. Lateral roots develop from a small subset of xylem pole pericycle (XPP) cells, designated as 'founder cells'. These founder cells then undergo a series of proliferative, asymmetrical cell divisions, establishing the new lateral root[39]. Our lateral root recorder was designed to track the expression of two genes: *ARABIDOPSIS HISTIDINE PHOSPHOTRANSFER PROTEIN 6* (*AHP6*) and *GATA TRANSCRIPTION FACTOR 23* (*GATA23*). AHP6 is a negative regulator of cytokinin signaling, and it plays an important role in guiding protoxylem formation[40] and orienting cell divisions during lateral root initiation. It is expressed in the xylem poles and XPP cells[41]. *GATA23* is strongly expressed in lateral root founder cells, setting into motion the series of cell divisions that form the lateral root[42]. For the integrase construct component of the lateral root recorder, we placed the first integrase (PhiC31) under the control of pAHP6 and the second integrase (Bxb1) under the control of pGATA23.

**Table 1 | Summary overview of integrase lines and characterization categories**

|  | As expected | Underswitched | Overswitched |
|---|---|---|---|
| pAHP6 single switch | RFP in xylem and lateral roots BFP otherwise | RFP in fewer cells than xylem and LR | RFP in more cells than xylem and LR |
| pGATA23 single switch | RFP in LR BFP otherwise | RFP in fewer cells than LR | RFP in more cells than LR |
| Lateral root history-dependent tracker | GFP in LR, RFP in xylem, BFP otherwise | GFP in fewer cells than LR and/or RFP in fewer cells than xylem | GFP in more cells than LR and/or RFP in more cells than xylem and LR |
| pSPCH single switch | RFP in guard cells and surrounding epidermal cells, BFP otherwise | RFP in fewer cells than guard cells and surrounding epidermal cells | RFP in more cells than guard cells and surrounding epidermal cells |
| pMUTE single switch | RFP in guard cellsBFP otherwise | RFP in fewer cells than guard cells | RFP in more cells than guard cells |
| Stomatal history-dependent tracker | GFP in guard cells RFP surrounding epidermal cells, BFP otherwise | GFP in fewer cells than guard cells and/or RFP in fewer cells than surrounding epidermal cells | GFP in more cells than guard cells and/or RFP in more cells than surrounding epidermal cells |

First, we characterized the single switches for pAHP6 and pGATA23. To characterize the performance of the single switches, we categorized seedlings as (1) 'as expected', meaning the only RFP-expressing cells were those that expressed the recorded gene (or descended from such a cell); (2) over-switched, meaning additional cells were expressing RFP; and (3) under switched meaning that fewer cells were expressing RFP (see Table 1 for a streamlined overview of these categories for each single integrase switch and full recorder and Supplementary Table 1 for more details on each construct). To note, in our system overswitching corresponds to switching happening at basal expression levels, meaning the integrase switch threshold is too low. Underswitching corresponds to an integrase switch threshold that is too high resulting in fewer cells than expected undergoing the switch. For our single switches and recorder lines, we characterized all T2 seedlings without selection for integrase constructs or homozygosity. Therefore, we observed unswitched T2 seedlings with all cells expressing BFP. We attributed this to the loss of the integrase construct between generations, as the proportion of these unswitched seedlings is consistent with that for prior integrase switches[22] and is consistent with Mendelian heritability (around 25%)[43], so we omitted these seedlings from our characterization.

For pAHP6, we transformed the 'PhiC31 then Bxb1' target line with a single integrase (PhiC31) under the control of the pAHP6 promoter, which we called the pAHP6 single integrase switch, as it only uses the switch from State 0 to State 1 of the target (Fig. 2a). In any cell where *AHP6* has been expressed, that is, cells in the xylem poles and XPP, we expected to see a switch from BFP to RFP expression, indicating a switch of the target from State 0 to State 1. Because the switch is heritable, we would predict that in a plant with this single pAHP6 switch, all xylem pole cells, XPP cells, and their descendants (e.g., lateral roots) would express RFP, representing the 'as expected' switch pattern (Table 1). Because no switch was observed initially in the xylem poles or XPP in any T1 seedlings, we added an NLS onto PhiC31 to increase its DNA recombination activity, resulting in four out of seven T1 seedlings with the 'as expected' switch pattern (Supplementary Table 2). Using this integrase construct with the NLS, we were able to consistently generate 'as expected' T2 seedlings (Fig. 2a). In the best performing pAHP6 single integrase switch T1 line (with NLS), 78% of T2 seedlings showed the expected switch pattern and the worst performing line had 18% with the expected pattern (Fig. 2a and Supplementary Fig. 4a). Part of this seedling to seedling variability is due to the fact that we characterized a mix of hetero and homozygous seedlings for the integrase construct. There were also seedlings consistently in the underswitched category (the line with the highest prevalence had 59% underswitched), in this case meaning the switch occurred in the lateral root but not in the xylem poles or XPP cells. This indicates stronger expression of *AHP6* in lateral root precursor cells, consistent with the fact that *AHP6* expression is induced by the plant hormone auxin[41], which is found at high levels in founder cells during early lateral root initiation[44].

*GATA23* has been previously used to drive a single PhiC31 integrase switch as a recorder of lateral root initiation[22]. To characterize a Bxb1-mediated single switch, we transformed the 'Bxb1 then PhiC31' target (Supplementary Fig. 3) with a single Bxb1 construct driven by pGATA23 such that Bxb1 mediates the switch from State 0 to 1 (Fig. 2b). As only one switch of the target is utilized, we refer to this as the pGATA23 single integrase switch. Consistent with our knowledge of *GATA23* expression and previously characterized *GATA23* integrase switch[22], we expected to observe RFP expression only in cells within the lateral root, constituting the 'as expected' switch category (Fig. 2b and Table 1). The pGATA23::Bxb1 single integrase switch resulted in T1 seedlings with RFP expression in cells outside the lateral root (overswitched) (Supplementary Table 2). To achieve switch specificity consistently in the T1 and T2 generations, we added a DST to the Bxb1 construct[22]. This pGATA23::Bxb1-DST single integrase switch resulted in five out of eight T1 seedlings with the 'as expected' switch pattern (Supplementary Table 2). From these 'as expected' T1s, we obtained three T1 lines out of five with T2 seedlings behaving 'as expected' (ranging from 15% to 71% of seedlings) (Fig. 2b and Supplementary Fig. 4). The majority of the T2 seedlings in most of the lines were overswitched. The T1 line with the lowest proportion of overswitching was 29%, and the highest line was 100% overswitched. The full T2 characterization of the single switches for the lateral root genes can be found in Supplementary Fig. 4a.

Next, we generated the full lateral root recorder, building the dual integrase construct with pAHP6-driven PhiC31 and pGATA23-driven Bxb1 and transforming it into the 'PhiC31 then Bxb1' target line. We added an NLS to the pAHP6::PhiC31 construct and no tag for the pGATA23::Bxb1 construct. According to the known expression patterns of *AHP6* and *GATA23*, we expected to observe xylem pole and XPP cells expressing RFP, lateral root cells expressing GFP, and the rest of the cells in the root should remain in their initial, BFP-expressing state (Fig. 2c). We were able to consistently generate seedlings with the 'as expected' pattern (Fig. 2d). In the T1 generation (Supplementary Table 3), four out of nine seedlings were over switched with the rest being some combination of 'as expected' and under switched. In the context of the full lateral root recorder, over-switched refers to seedlings with PhiC31 and/or Bxb1 over-switching, such as with GFP-expressing cells outside the lateral root and/or RFP-expressing cells outside the xylem poles and XPP (Supplementary Fig. 5).

We characterized T1 lines from the 'as expected' and under-switched T1s because subsequent generations are likely to have stronger integrase expression due to higher copy number in homozygous individuals. In the T2 generation, an average of 23% of seedlings screened per line showed the expected output. Those that deviated from the expected phenotype consisted primarily of overswitched seedlings (Supplementary Fig. 6a). In contrast to the pGATA23 single switch where the pGATA23::Bxb1 construct led to overswitching in every T1 seedling (Supplementary Table 2), in the full recorder we were able to achieve switching specific to *GATA23* expression without

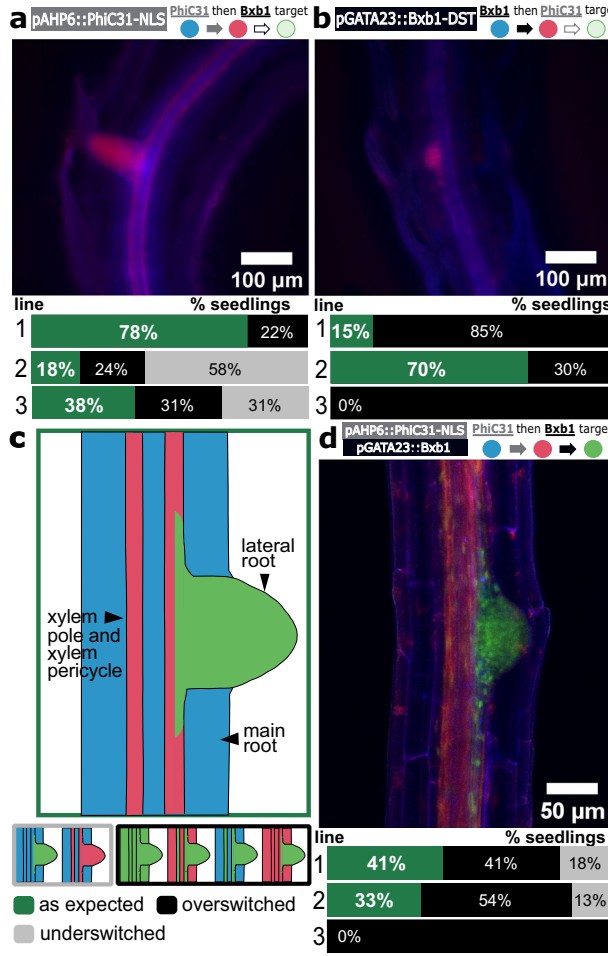

**Fig. 2 | Application of the history-dependent recorder in lateral roots. a** A pAHP6 single integrase switch. Plants carrying the 'PhiC31 then Bxb1' target were transformed with pAHP6::PhiC31-NLS. A representative image of a T2 seedling is shown with the switch characterization for three representative lines (line 1: T2P5 (*n* = 17), 2: T2P2 (*n* = 16), 3: T2P7 (*n* = 9)), see Supplementary Fig. 4a), where n represents the number of seedlings characterized in each line, omitting seedlings with no switching. The bar graphs represent the proportion of seedlings in each T1 line, showing the 'as expected', overswitched, and underswitched switch pattern (as defined in Table 1). **b** A pGATA23 single integrase switch. Plants carrying the 'Bxb1 then PhiC31' target (Supplementary Fig. 3) were transformed with pGATA23::Bxb1-DST. A representative image of a T2 seedling is shown with the switch characterization for three representative lines (line 1: T2P5 (*n* = 17), 2: T2P3 (*n* = 20), 3:T2P1 (*n* = 16), see Supplementary Fig. 4a). The bar graphs represent the proportion of seedlings in each T1 line showing the 'as expected', over switched, and under switched switch pattern (as defined in Table 1). **c** Predicted recorder output. Xylem pole and xylem pole pericycle cells should express RFP (State 1, red), the cells in the lateral root should express GFP (State 2, green), and any other cell should remain in the initial BFP-expressing (State 0, blue). Below are schematic representations of the underswitched (left) and overswitched (right) patterns (see Supplementary Fig. 5 for more detail). **d** Lateral root recorder output. T2 seedlings from 5 lines, which came from T1 plants with an 'as expected' or under-switched recorder output, were screened. A representative image for one of these T2 seedlings is shown, as well as a characterization of the recorder output for three representative T1 lines (line 1: T2P1 (*n* = 17), 2: T2P3 (*n* = 15), 3: T2P9 (*n* = 13), see Supplementary Fig. 6a). The bar graphs represent the proportion of seedlings in each T1 line showing the 'as expected', over switched, and under switched switch pattern (as defined in Table 1). Source data are provided as a Source Data file.

addition of a DST to Bxb1. In our dual integrase construct, the Bxb1 transcriptional unit is immediately downstream of the PhiC31 unit, and this construct architecture has been shown to result in reduced expression of the downstream gene[45], possibly due to transcription-induced DNA supercoiling[46]. We observed negligible intra-plant

variation in the fluorescent output of the *AHP6* single integrase switch, the *GATA23* single integrase switch, and the full lateral root recorder. Lateral roots of similar developmental stages on the same seedling showed the same fluorescent output, indicating low levels of intrinsic noise in the integrase switches. Schematics of all the observed switch patterns can be seen in Supplementary Fig. 6. Overall, we implemented a recorder tracking the expression patterns of AHP6 and GATA23 during lateral root initiation and consistently generated T2 seedlings with the expected switch pattern.

## History-dependent tracking of gene expression during stomatal development

We next applied our history-dependent recorder to a different well-characterized differentiation program: stomatal development. Stomata are each composed of two guard cells which together form a small pore in the leaf surface. These pores are required for efficient gas exchange and are a target of interest for engineering increased carbon capture with higher water use efficiency[47].

Each stoma descends from a proliferating meristemoid cell on the leaf epidermis, which serves as the precursor of the stomatal cell lineage. Stomata develop following a stereotypical progression wherein the meristemoid undergoes some number of asymmetrical cell divisions before dividing symmetrically exactly once into the two guard cells that comprise the stoma. Many of the genes involved in this progression from meristemoid to guard cell have been extensively studied[48]. We focused on recording the expression of two genes in the context of stomatal development: *SPEECHLESS* (*SPCH*), which enables the asymmetrical cell divisions of the meristemoid, and *MUTE*, which is expressed later and is essential to trigger the single symmetric division into two guard cells[49]. We, therefore, used pSPCH to drive the expression of the first integrase (PhiC31) and pMUTE to drive the expression of the second integrase (Bxb1). The final recorder line contains this integrase construct and the 'PhiC31 then Bxb1' target.

We first separately tested the individual integrase switches with pSPCH driving PhiC31 and pMUTE driving Bxb1. For the pSPCH single switch, we transformed the pSPCH-driven PhiC31 construct into the 'PhiC31 then Bxb1' target line, expecting to observe a switch from BFP to RFP expression (State 0 to 1) in the guard cells plus any surrounding epidermal cells that resulted from asymmetric meristemoid divisions (the 'as expected' switch category) (Fig. 3a and Table 1). In this pSPCH single integrase switch, over-switched means that additional leaf epidermal cells were expressing RFP and under-switched means that fewer cells were expressing RFP (Table 1). In the T1 generation, four out of the ten seedlings showed 'as expected' switching (Supplementary Table 4). In the T2 generation, we obtained up to 50% of seedlings per line switching 'as expected' with a high variability between lines. Most of the T2 seedlings showed an overswitched phenotype, with some or all leaf epidermal cells not in contact with stomata expressing RFP. The most over-switched T1 line had 100% over-switched seedlings, while the lowest rate of overswitching observed was 14%. One line had 64% of the seedlings under-switched, with only the guard cells of the stomata expressing RFP (Fig. 3a and Supplementary Fig. 4b).

To generate the pMUTE single switch, we built a pMUTE-driven Bxb1 construct and transformed it into the 'Bxb1 then PhiC31' target (Supplementary Fig. 3) line. We characterized the lines similarly to the pSPCH single switch lines in the T1 and T2 generations, with the 'as expected' phenotype being RFP expression in the guard cells of the stomata only (Table 1). In the T1 generation, five out of ten seedlings showed the 'as expected' switch pattern (Supplementary Table 4). In the T2 generation, we consistently obtained a majority of seedlings expressing RFP only in the guard cells (73% and 53%) (Fig. 3b). Every seedling that did not fit this pattern showed an over-switched phenotype, with RFP expression in additional leaf epidermal cells (Fig. 3b and Supplementary Fig. 4b). For both the single integrase switches, we observed negligible variation in the switch pattern across locations on

the leaf, with leaf epidermal cells throughout showing a consistent switch output, whether it was under switched, over switched, or 'as expected'.

We then generated the full stomatal recorder by building the dual integrase construct with pSPCH-driven PhiC31 and pMUTE-driven Bxb1 and transforming it into the 'PhiC31 then Bxb1' target line. We expected to see expression of GFP in the guard cells of the stomata, expression of RFP in any surrounding epidermal cells that resulted from asymmetric meristemoid divisions, and expression of BFP in the remaining leaf epidermal cells that are not involved in stomatal differentiation. Of the nine T1 seedlings generated, six showed GFP expression specific to stomata, and the lines from these six seedlings were characterized (Supplementary Table 3). In five out of the six lines, we obtained seedlings with the expected phenotype (Fig. 3d and Supplementary Fig. 6b), with a proportion of up to 50%, and a low of 14% 'as expected' seedlings per line, with three out of six lines having an 'as expected' proportion greater than 33%. The majority of the not 'as expected' seedlings were over-switched, meaning they were expressing GFP in cells other than the guard cells and/or expressing RFP in excess epidermal cells. The over-switched category accounted for the majority of seedlings in most of the stomatal recorder T1 lines, including 100% of seedlings in one line, with a low of 35% and a median of 68% of seedlings. Most of the overswitched seedlings (median of 67%) showed both PhiC31 and Bxb1 overswitching (Supplementary Fig. 7). A small proportion of seedlings (15% or less) showed PhiC31 underswitching, meaning the guard cells were expressing GFP and the rest of the epidermal cells were expressing BFP, but no cells were actively expressing RFP. For one line, we obtained only seedlings with non-specific switching (both PhiC31 and Bxb1 over switching), meaning some or all non-stomata epidermal cells were expressing GFP (switched to State 2) and/or some or all epidermal cells not in contact with at least one stoma were expressing RFP (switched to DNA State 1). A schematic summary of all the observed recorder outputs is in Supplementary Fig. 7. As highlighted previously, some of the variability between seedlings is due to differences in the zygosity of the integrase construct. For the stomatal recorder, we performed characterization of homozygous and non-homozygous T3 lines and found the homozygous line to be significantly more overswitched, likely due to higher integrase expression (Supplementary Fig. 8). Despite the observed variability, we consistently generated seedlings in the 'as expected' category for the stomata recorder and the *SPCH* and *MUTE* single integrase switches.

## The history-dependent recorder can identify cells that have undergone an alternate developmental path

For the recorder seedlings with an 'as expected' phenotype, the majority of the stomata expressed GFP as expected; however, there was a subset of stomata that did not express any fluorescent protein (Fig. 4a). In the design of the recorder, a cell which has no fluorescence indicates that the target has been switched to State α due to Bxb1 recombination before PhiC31 recombination. This result suggests the possibility that there are two stomatal populations that differ in how likely they are to switch to State α versus to proceed through the series of switches to State 2. We did not observe such variation in the single *SPCH* and *MUTE* integrase switches, so we hypothesized that this discrepancy was due to differences in the relative timing of *SPCH* and *MUTE* expression. A closer relative timing of expression would increase the chances of Bxb1 mediating the switch to State α, while a longer timing would give PhiC31 more of a head start, increasing the likelihood of the cell switching to State 1 and then State 2. Indeed, temporal logic gates based on similar integrase designs have been employed previously in *E. coli* to reveal the timing between events[50].

It has been shown that the timing between *SPCH* and *MUTE* expression differs between the development of two types of stomata: anisocytic (A) and nonanisocytic (NA)[51]. During stomatal

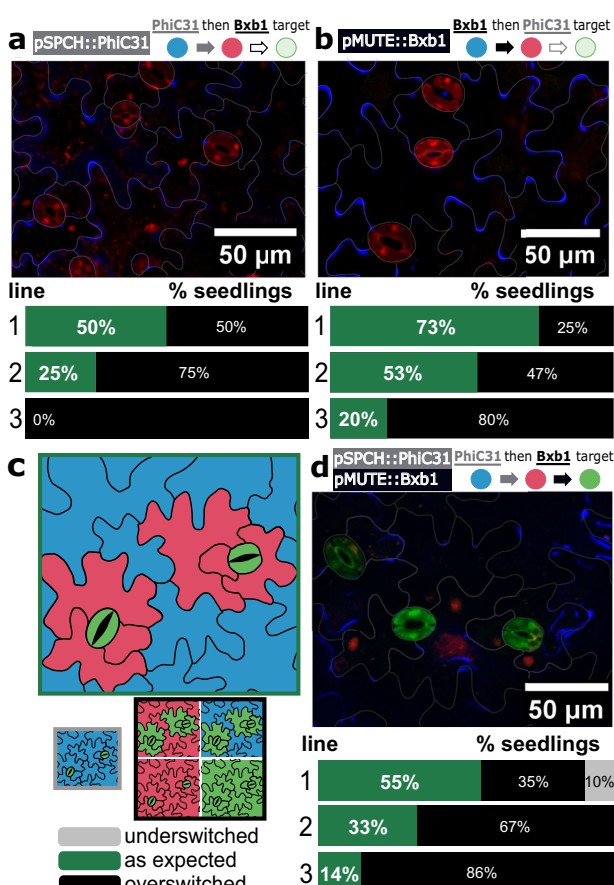

**Fig. 3 | Application of the history-dependent recorder during stomata development. a** A pSPCH single integrase switch. Plants carrying the 'PhiC31 then Bxb1' target were transformed with pSPCH::PhiC31. A representative image of a T2 seedling is shown, as well as the switch characterization for three representative lines (line 1: T2P3 ($n = 12$), 2: T2P8 ($n = 14$), 3: T2P1 ($n = 14$), see Supplementary Fig. 4b) where n represents the number of seedlings characterized in each line omitting seedlings with no switching. **b** A pMUTE single integrase switch. Plants carrying the target were transformed with pMUTE::Bxb1. A representative image of a T2 seedling is shown with the switch characterization for three representative lines (line 1: T2P2 ($n = 15$), 2: T2P7 ($n = 15$), 3: T2P1 ($n = 15$), see Supplementary Fig. 4b). **c** Prediction of stomatal recorder output. Guard cells are predicted to be in State 2 and expressing GFP (State 2, green), surrounding epidermal cells that are the result of meristemoid division should be expressing RFP (State 1, red), and any other cell should sustain the initial BFP expression (State 0, blue) Below are schematic representations of the underswitched (left) and overswitched (right) patterns (see Supplementary Fig. 7 for more detail). **d** Experimental stomatal development recorder output. The 'PhiC31 then Bxb1' target is transformed with the dual stomatal integrase construct. One representative image of a T2 seedling matching the expected output is shown. The recorder performance is shown for three representative lines (line 1: T2P1 ($n = 20$), 2: T2P4 ($n = 18$), 3: T2P2 ($n = 20$), see Supplementary Fig. 6b), with percentages indicating the proportion of seedlings matching the expected recorder output and n representing the number of seedlings characterized in each T1 line omitting seedlings with no switching. Source data are provided as a Source Data file.

differentiation, *SPCH* initiates, and *MUTE* terminates the asymmetrical divisions of the meristemoid[49]. Closer timing of expression between *SPCH* and *MUTE* permits fewer asymmetrical divisions, resulting in an NA stoma, whose development can include zero, one, or two asymmetric meristemoid divisions. In contrast, a longer delay between *SPCH* and *MUTE* expression allows the full three asymmetrical cell divisions needed for development of an A stoma (Fig. 4b). As a result, A stomata, the majority of stomata in most cases, are easily identified because the series of three asymmetrical divisions results in a stoma

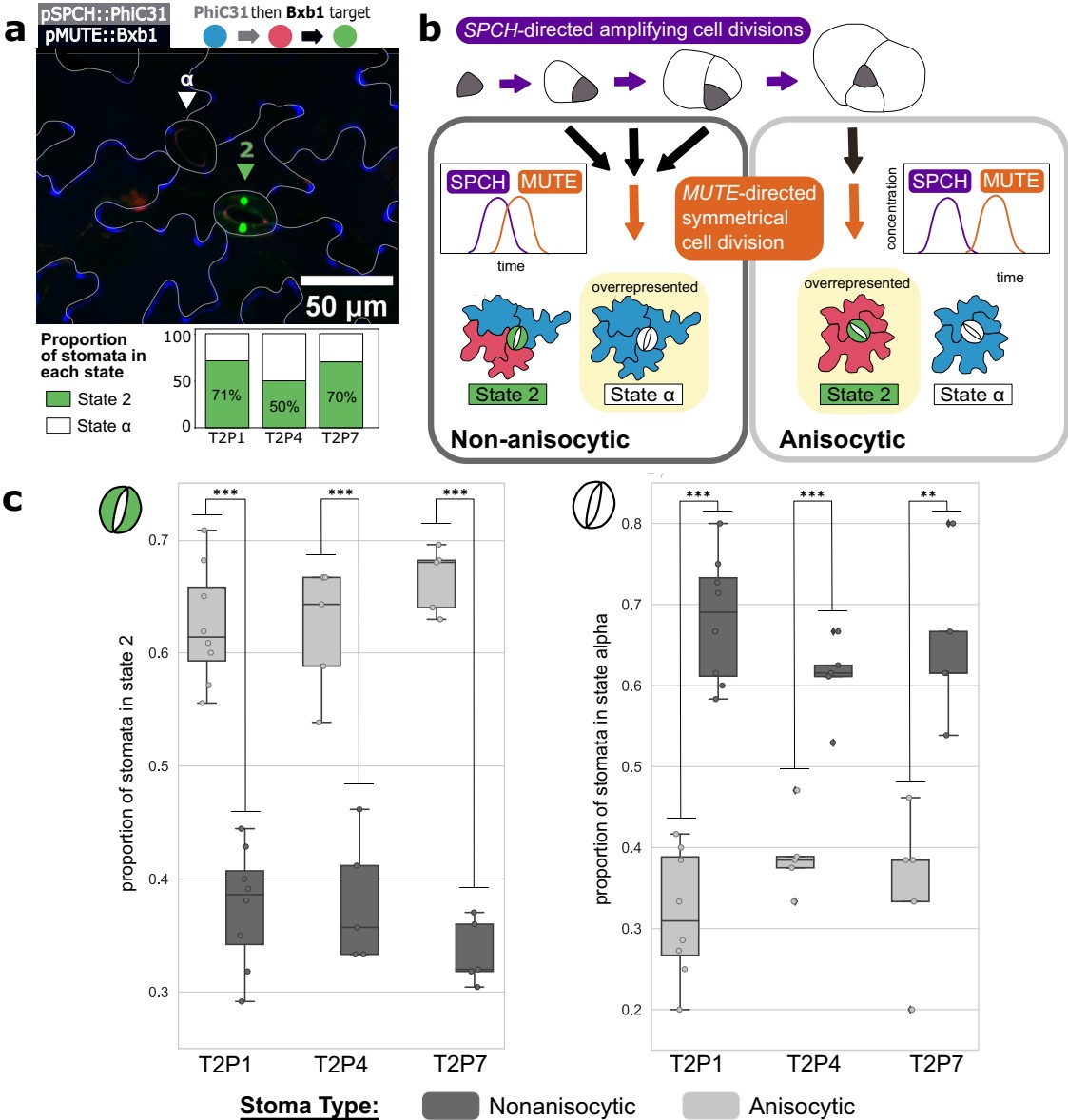

**Fig. 4 | The history-dependent recorder reveals two distinct stomatal populations. a** State 2 and State α stomata. A confocal image shows two stomata: one in State 2 (green triangle) and one in State α (white triangle). Below the image, the total proportion of stomata in each state is shown in a barplot, with each bar representing one of the three best-performing stomata recorder lines (labeled on the x-axis). $n = 257$, $n = 172$, and $n = 174$ stomata were characterized for T2P1, T2P4, and T2P7. **b** Alternate paths of stomatal development. The stereotypical understanding of stomatal development follows an anisocytic (A) scheme, with the meristemoid dividing asymmetrically three times before dividing into two guard cells. Non-anisocytic (NA) stomata divide asymmetrically zero, one, or two times before the terminal division. Recorder output is predicted to differ between stomata types due to varied timing of integrase onset. An A stomata is surrounded by exactly three cells of unequal size, while an NA stomata can have two, four, five, or three equally sized neighbor cells. **c** Characterization of recorder state in A vs. NA stomata. The boxes represent the proportion of each stoma type (anisocytic in light gray and non-anisocytic in dark gray) within each of the three stomata recorder lines (x-axis) that are in State 2 (left) and State α (right). Each box represents the middle quartiles of the respective dataset, and the whiskers represent the highest and lowest quartiles. The center of each box is the median. Each point represents the proportion of stomata in each state for one T2 seedling. 10 seedlings were counted per line. For each seedling, a minimum of 15 of each stomata type was counted (at least 30 total stomata per seedling). To evaluate the statistical significance between the result for A and NA stomata populations, a two-sided Student's $t$ test was performed (*$p < 0.05$, **$p < 0.01$, ***$p < 0.001$). From left to right in the plots, $p$-values are $p = 0$, $p = 0.0001$, $p = 0$, $p = 0$, $p = 0.0001$, $p = 0.001$. Source data are provided as a Source Data file.

surrounded by exactly three cells of unequal size, whereas an NA stomata can have two, four, or five neighbor cells or three neighbor cells of roughly equal size[52] (Fig. 4b).

We hypothesized these two developmental lineages, with their differences in relative gene expression timing, led to two stomata populations with different proportions of cells in each recorder state. We predicted that the State 2 stomata should be overrepresented in the A population and State α stomata should be overrepresented in the NA population. To test this hypothesis, we counted 25 stomata per

T2 seedling in 10 individuals. This was repeated for the three best-performing stomata recorder lines, only characterizing seedlings matching the 'as expected' phenotype (Supplementary Fig. 7). For each counted stoma, we categorized it as A or NA and State 2 or State α (corresponding to GFP expression or no expression, respectively). We then calculated for both stomatal populations, the percentage of stomata in State 2 versus in State α. For A stomata, an average of 65% across all three lines were in State 2 and 35% in State α. In contrast, for NA stomata, an average of 33% across the three lines were in State 2,

with 67% in State α (Fig. 4c). Therefore, as predicted, State 2 stomata were enriched in the A population compared to NA where a majority of stomata were in State α. In theory, the State α stomata could be a result of stomatal differentiation, which skips *SPCH* expression entirely. If this were the case, the pSPCH single integrase switch result (Fig. 3a) would include unswitched stomata at a comparable rate, however we observed that all stomata in the 'as expected' pSPCH single integrase switch lines expressed RFP and therefore include *SPCH* expression in their lineage. This result supports our hypothesis that the disparity in recorder output was split probabilistically based on the extent of asymmetrical cell divisions prior to *MUTE* expression. Overall, we showed that the state of the recorder in a given stoma correlated strongly with its developmental history and stomatal type. The successful encoding of differences in the relative timing of *SPCH* and *MUTE* expression during alternative developmental trajectories demonstrates the potential utility of this strategy in connecting patterns of gene expression to differentiation in many other contexts.

## Discussion

Developmental biology has made enormous strides in identifying gene regulatory networks that specify cell fate at a population scale (i.e., for the 'average' cell with a given identity). There is increasing interest in understanding the history of individual cells, including quantifying cell-to-cell variation in gene expression dynamics and context-specific, cell non-autonomous influences on cell fate. We have built a tool that can help with these efforts. Our integrase-based history-dependent recorders worked in two distinct developmental contexts: the initiation of a new root from the pericycle cells of the primary root and the differentiation of stomata in the leaf epidermis. In both cases, we could readily visualize cells in State 0, State 1, and State 2, and we consistently generated seedlings whose fluorescent expression matched our expectations based on known gene expression patterns. In tracking stomatal development, we also detected guard cells with no fluorescence, indicating a switch to State α. This finding supports prior evidence for the existence of two populations of stomata. Our results supported the hypothesis that the difference between stomata in State 2 and State α reflect differences in the relative time between the expression of the two genes of interest. Collectively, our work demonstrates the utility of the recorder for revealing variation in the timing of gene expression events even in seemingly uniform cell differentiation processes.

For a targeted study of the gene expression patterns underlying cell development paths of interest, our recorder provides an accessible and powerful methodology. Our recorder was sensitive enough to capture a disparity in gene expression timing in a very well-studied context, so it should prove even more beneficial for those working in less-studied cell development contexts. This sensitivity, combined with the strong constitutive, sustained expression of the reporters, allows the recorder to act as an amplifier of the gene expression signal, which should prove useful for studying low-expressing genes not suited for traditional transcriptional reporters and enables evaluation of developmental events over timescales far exceeding the degradation time of the fluorescent proteins. While we applied our recorder design to well-characterized differentiation processes, it can be adapted to processes wherein the patterns of gene expression are not well understood. This would necessitate additional optimization of the integrase switch specificity to reduce plant-to-plant and cell-to-cell variability combined with gene expression validation using a transcriptional reporter or spatial transcriptomics data. With this extra optimization and validation step, our recorder offers unique advantages for less well-studied genes, including sustained fluorescent expression to enable recording even when the exact timing of expression is unknown; high sensitivity to enable recording even at low levels of expression; the ability to discern variation from cell to cell; and the tracing of the cell lineage which had expressed the gene of interest. Our recorder enables a snapshot readout of each cell's gene expression history with only standard molecular biology techniques and access to a fluorescent microscope to take images at just a single or small number of time points.

While much information could be gleaned from the recorders analyzed here, future engineering efforts, especially those focusing on less well-characterized pathways, may require a systematic approach to reduce variability. This issue is a persistent one for engineering in plants. Line-to-line variation is likely due to the random insertion location of transgenes leading to different levels of gene expression. This issue could be addressed using targeted DNA transformation techniques. Variability in performance between seedlings carrying the same transgene insertions could be addressed with iterative rounds of optimization with our previously characterized tuning parts[22]. In the future, information from a thorough characterization of integrase efficiency combined with knowledge of gene expression strength could inform the choice of tuning parts, expediting the process of generating robust and specific integrase switches. In this study, variability is exacerbated by varied integrase expression levels among seedlings within T1 lines due to differences in the zygosity of the integrase construct. For future applications, characterization and optimization of T3 homozygous lines will be required.

A four-state recorder like the designs used here can capture the history of expression for two genes. Using our current target design, researchers could generate multiple recorders within the same cell specification pathway to capture the expression patterns for more gene combinations. Moreover, the current recorder infrastructure could be adapted following existing integrase target designs to generate a target with a higher number of inputs[25]. The design capabilities of integrase circuits[23] combined with the large number of characterized orthogonal integrases[27,53] should enable the construction of circuits with higher numbers of outputs. The possible scope is limited in principle only by the availability of spectrally distinct reporters, and this limit could be overcome with multiplexing techniques[54] or switching to DNA barcodes[20].

In theory, our recorder designs should be transferable to any plant that can be transformed, for applications such as those facilitating evo-devo studies of key developmental processes such as stomata. Our recorder can be applied in plants that undergo different paths of stomatal development, such as in amphibious plants which, largely skip the *SPCH*-driven, asymmetrical cell divisions[55], and grasses which, have a completely different arrangement of the subsidiary cells surrounding the stomata compared to *Arabidopsis*[56]. In addition, integrase switches can be adapted to allow the study of essential genes in non-essential organs[57]. Porting this system into crop plants could enable optimization and quantification of traits associated with climate resilience, facilitating marker-assisted breeding efforts. Our history-dependent recorder should also function across different non-plant organisms and should be applicable to challenges such as studying differentiation paths involving lowly-expressed genes or evaluating cell-to-cell variability in gene expression trajectory. In applications beyond tracking gene expression, integrase circuits like the ones described here could be modified for engineering applications. If the reporters were replaced with developmental genes and integrase expression was tied to chemical signals, specific stress responses, or specific tissues– interactive synthetic genetic programming of development should be entirely feasible for applications such as programming climate-resilient root architectures. The relative ease and accessibility of our method should enable rapid prototyping and optimization for use in elucidating and reprogramming cell states.

## Methods
### Construction of plasmids
Our cloning strategy was based on Golden Gate assembly using an appropriate spacer and BsaI-HFv2 (NEB) and BbsI-HF (NEB) as the

restriction enzymes (Supplementary Fig. 9). Details on all primers and constructs can be found in Supplementary Datas 1 and 2 respectively. Candidate promoter sequences (SPCH: AT5G53210, MUTE: AT3G06120, AHP6: AT1G80100, GATA23: AT5G26930) were amplified from Col-0 genomic DNA (primer list available in Supplementary Data 1) to add specific Golden Gate spacers. After PCR purification, each level 0 promoter sequence was cloned using a Zero Blunt PCR Cloning Kit (ThermoFisher Scientific). The PhiC31 integrase sequence was a gift from the Orzeaz lab. The Bxb1 sequence was a gift from the Bonnet lab. Integrases were amplified using primers with golden gate compatible spacers to generate level 0 integrase parts. Constitutive plant promoters and terminators were purchased from Addgene as part of the MoClo Toolbox for Plants[58]. A mutated version of the pPP2AA3 promoter without BsaI sites was ordered from Twist Bioscience. Other level 0 fragments were ordered from IDT as Gblocks: the fluorescent proteins NeonGreen-NLS and mtagBFP2-ER and combinations of integrase sites and restriction sites for the construction of various integrase targets. The mCherry sequence was amplified from a constitutively expressed mCherry construct (donated by Jennifer Brophy). For the integrase target level 0 sequences, the p35S or pUBQ10 promoter was added by Golden Gate using BbsI sites (Supplementary Fig. 9).

The construction of level 1 single integrase constructs (constitutive, stomata, and lateral root specific) was performed via Golden Gate reaction in the modified pGreenII-Hygr vector containing compatible Golden Gate sites[58]. The construction of integrase targets was performed with the same methods in a modified pGreenII-Kan vector. Construction of level 2 integrase constructs expressing both PhiC31 and Bxb1 for the full recorders was performed by amplifying completed level 1 integrase constructs using primers with Golden Gate compatible spacers, then performing Golden Gate reactions in the modified pGreenII-Hygr vector containing compatible Golden Gate sites. The addition of an NLS or DST to level 1 and level 2 integrase constructs was performed using PCR-mediated site-directed mutagenesis (NEB Q5 Site-Directed Mutagenesis, Cat #E0554S). Primers for mutagenesis were designed with NEBasechanger (primer list available in Supplementary Data 2). Enzymes for Golden Gate assembly were purchased from New England Biolabs (NEB, Ipswich, MA, USA). PCR was performed using a 2X Q5 PCR master mix (NEB) and a GoTaq master mix for colony PCR and genotyping (Promega, Madison, WI, USA). Primers were purchased from IDT (Coralville, IA, USA), and DNA fragments from Twist Bioscience (San Francisco, CA, USA) or IDT. Plasmid extraction and DNA purification were performed using Monarch kits (NEB). Sequences were verified with Sanger sequencing by Azenta Life Sciences or Primordium Labs for whole plasmid sequencing. Chemically-competent cultures of the *E. coli* strain DH5αZ1 (laciq, PN25-tetR, SpR, deoR, supE44, Delta(lacZYA-argFV169), Phi80 lacZ-DeltaM15, hsdR17(rK − , mK + ), recA1, endA1, gyrA96, thi-1, relA1) were transformed with plasmid constructs containing kanamycin resistance. Transformed *E. coli* was grown in LB media (LB broth, Miller) with kanamycin (Millipore Sigma, 50 μg/mL).

## Plant growth conditions
*Arabidopsis* seedlings were sown in 0.5 x Linsmaier and Skoog nutrient medium (LS) (Caisson Laboratories) and 0.8% w/v Phyto agar (PlantMedia/bioWORLD), stratified at 4 °C for 2 days, and grown in constant light at 22 °C.

## Construction and selection of transgenic *Arabidopsis* lines
*Agrobacterium tumefaciens* strain GV3101 was transformed by electroporation, and subsequently grown in LB media with rifampin (Millipore Sigma, 50 μg/mL), gentamicin (Millipore Sigma, 50 μg/mL), and kanamycin (Millipore Sigma, 50 μg/mL). The floral dip method[59] was used to generate integrase target lines in Col-0, and then transformation of the integrase constructs into the target lines was performed

via an adjusted dual dipping approach wherein plants are dipped on two occasions approximately a week apart (with the first dipping occurring when the stems are around 3 inches tall) and the agrobacterium were incubated with MMA (10 mM MgCl2, 10 mM MES pH 5.6, 100 μM acetosyringone) for an hour prior to dipping, as acetosyringone is known to improve agrobacterium-mediated transformation efficiency in *Arabidopsis* explants[60]. For T1 selection: 150 mg of T1 seeds (~2500 seeds) were sterilized using 70% ethanol and 0.05% Triton-X-100 and then washed using 95% ethanol. Seeds were resuspended in 0.1% agarose and spread onto 0.5X LS Phyto selection plates, using 25 μg/mL of kanamycin for target lines or 25 μg/mL kanamycin and 25 μg/mL hygromycin for lines with both the integrase construct and the target. The plates were stratified at 4 °C for 48 h and then grown for 7, 8 days. To select transformants, tall seedlings with long roots and vibrant green color were picked from the selection plate with sterilized tweezers and transferred to a new 0.5X LS Phyto agar plate for characterization.

## Plant genotyping
For the genotyping of plants, leaves from approximately 20 day old plants were used. For plant DNA extraction, leaf sections approximately 1 cm² in area were frozen on dry ice and ground for 1 min into a fine powder. 10 μm 0.5 M NaOH was added to each sample before boiling them for 1 min and then adding 100 μL neutralization solution (1 part 1 M Tris-OH/HCl, pH 8.0 and 4 parts 0.1 M TE, pH 8.0). Genotyping PCRs were set up using the GoTaq master mix (Promega, Madison, WI, USA), making sure to gently stir the DNA extraction with the pipette tip before pipetting into the reaction.

## Hygromycin selection to determine integrase zygosity in T3 lines
T3 seeds were plated on 0.5 X LS and 0.8% w/v Phyto agar with 50 μg/mL hygromycin. Seeds were then stratified at 4 °C for 2 days, and then grown in constant light at 22 °C for 10 days before hygromycin resistance was evaluated. Seedlings with substantial root growth were characterized as resistant, while those with extremely stunted root growth were considered non-resistant.

## Imaging of reporter and integrase lines
Initial screening of root and leaf tissue was performed using a Leica Biosystems microscope (model: DMI 3000) with a 10x objective for imaging roots and a 40x objective for imaging leaf tissue. Further characterization and images in the manuscript were obtained via confocal imaging.

Confocal imaging of the seedling root and leaf tissue was performed using a Nikon A1R HD25 laser scanning confocal microscope with a Plan Apochromat Lambda 20x objective. Three channels were used: 561 laser and 578-623 detector for RFP imaging; 488 laser and 503−545 detector for GFP imaging; 405 laser and 419−476 detector for BFP imaging. For each image, a Z-stack was recorded. All images were processed using ImageJ FIJI.

Confocal imaging of lateral root recorder was done after characterization of T1 seedlings and subsequent generation of T2 seeds. Ten- day old T2 seedlings were mounted on slides with water and with Parafilm edges to prevent the coverslip from pressing on the root. The main root of each seedling was scanned for pre-emergence lateral roots to image. The gain was set at 75, 45, and 40 respectively for the BFP, RFP, and GFP channels. The laser power was set to 3, 10, and 4, respectively.

Confocal imaging of stomata single switches and recorder was performed after the characterization of T1 seedlings and subsequent generation of T2 seeds. The first true leaves from 12-16-day-old T2 seedlings were cut off and mounted on slides with 50% glycerol for imaging of the abaxial side of the leaf. The edges of the coverslip were painted with clear nail polish to prevent movement of the sample

during imaging. The leaves were scanned to locate the flattest areas to take images. The gain was set at 75, 50, and 35, respectively, for the BFP, RFP, and GFP channels. The laser power was set to 2, 12, and 3, respectively.

## Characterization of the history-dependent recorder in Arabidopsis transgenic lines

T1 seedlings for each line were grown 4–5 days after transformant selection. Each selected seedling was imaged at 10x magnification using an epifluorescence microscope (Leica Biosystems, model: DMI 3000) using the GFP (exposure 400 ms, gain 2), RFP (exposure 500 ms, gain 2), and CFP (exposure 300 ms, gain 2) channels. Selected T1 seedlings were then transferred to soil, and at maturation, T2 seeds were collected. For later generations, seedlings were sterilized similarly to T1s, stratified, plated on an LS agar plate, and grown for either 10 days (for characterizing roots) or 12–16 days (for characterizing leaves). Target characterization was done using the epifluorescence microscope as for T1. For the target lines, the seedlings with the highest level of mtagBFP2 expression in the root (or, in the case of the pre-switched target, the highest level of mCherry expression) were selected and transferred to soil to generate T2 seeds. The line with the most consistently bright seedlings was maintained as the target line for each integrase target and used for all later transformations of integrase constructs.

For the constitutive integrase constructs in a target line, around 5-10 T1 seedlings were analyzed per construct, and the ones that displayed any level of switching were transplanted to the soil for characterization in the T2 generation, where around 20 seedlings were characterized per line. Representative root images were taken using the RFP, GFP, and CFP channels and merged for final images. Representative leaf images were generated using a Nikon A1R HD25 laser scanning confocal microscope as described above.

## Characterization in the context of lateral root development

For the pAHP6 and pGATA23 single integrase switches, at least 15 T1 seedlings were analyzed per construct. Each seedling was categorized into one of three classes based on the specificity of switching: 'as expected' (RFP expression in xylem pole, XPP, and lateral root cells); over switched (RFP expression in additional cells); under switched (RFP expression in fewer cells). The seedlings in the no switch category represented around 25% of seedlings and were omitted from analysis as this proportion is consistent with a loss of the integrase construct through gene segregation[21]. Representative images were taken in the epifluorescence microscope using the GFP, RFP, and CFP channels and merged for final images. For the *AHP6* switch, T1 seedlings, which showed switching in the xylem poles, xylem pericycle, and lateral root cells, were transplanted to the soil along with T1 seedlings which showed switching only in the lateral root. For the pGATA23 single integrase switch, only T1 seedlings that showed switching specific to the lateral root were transplanted for future T2 characterization. For each selected T1 line, 15-20 T2 seedlings were characterized, categorizing each as follows: 'as expected' (RFP expression in lateral root cells); overswitched (RFP expression in additional cells); underswitched (RFP expression in fewer cells) (Table 1).

For the full lateral root recorder to track both *AHP6* and *GATA23* expression, nine T1 seedlings were analyzed and those whose switch pattern was either 'as expected' or some form of underswitched (Supplementary Fig. 6), were transplanted to soil for future T2 characterization. From each T1 line, 15–20 seedlings were characterized and categorized based on switch pattern (Supplementary Fig. 5). For Fig. 2, these categories were consolidated into under-switched, 'as expected', and over-switched categories (Table 1) as for the single switches, with any seedlings with Bxb1 and/or PhiC31 over switching fitting into the over switched category and any seedlings with Bxb1 and/or PhiC31 under switching fitting into the under switched category (Supplementary Fig. 5). Representative images for the full lateral root

recorder were taken with the Nikon A1R HD25 laser scanning confocal microscope.

## Characterization in the context of stomata development

For the single pSPCH and pMUTE T1 single integrase switches, leaves from at least 10 T1 seedlings were analyzed per construct. Due to the chlorophyll autofluorescence causing difficulties in imaging mCherry in the epifluorescence microscope, this characterization was done using the confocal microscope. For the pSPCH single integrase switch, T1 seedlings with switching in only stomata and some epidermal cells that border stomata were transplanted for T2 characterization. T1 seedlings with switching limited to only stomata were also transplanted. For the pMUTE single integrase switch, seedlings with switching only in the stomata were transplanted for T2 characterization. For the full stomata recorder for tracking *SPCH* and *MUTE* expression, nine T1 seedlings (Supplementary Table 3) were characterized, and any with GFP expression specific to stomata were transplanted for T2 characterization.

For the pSPCH and pMUTE single integrase switch T1 lines, leaves from 15-20 seedlings were characterized as described for T1. The pSPCH single integrase switch seedlings were sorted into one of three categories: 'as expected' (RFP expression in guard cells and surrounding epidermal cells); overswitched (RFP expression in additional cells); underswitched (RFP expression in fewer cells). For the pMUTE single integrase switch, the categories are as follows: 'as expected' (RFP expression in only guard cells); overswitched (RFP expression in additional cells); underswitched (RFP expression in fewer cells). For the full recorder the categories are as seen in Supplementary Fig. 7. For Fig. 3 these categories were consolidated into the same underswitched, 'as expected', and overswitched categories as for the single switches, with seedlings with either or both PhiC31 and Bxb1 overswitching fitting into the overswitched category and seedlings with either PhiC31 or Bxb1 underswitching fitting into the underswitched category. Representative images for the single switches and the full recorder were taken with the Nikon A1R HD25 laser scanning confocal microscope.

For quantifying the relationship between recorder output and stomata type, leaves from seedlings were stained with 2 mg/mL Calcofluor White for 30 minutes (Millipore Sigma Cat #18909) to better visualize the cell boundaries. To screen random stomata all across the leaf, we started from the top left of each leaf and shifted the viewing frame of the microscope across the leaf incrementally and then down at least a full frame before moving back across the leaf. In each frame, the centermost stomata were chosen to be categorized. This categorization involved determining the type of stomata based on the pattern of surrounding cells and then noting the recorder output (GFP expressing or no fluorescence). In the case that the type of stomata was not able to be determined, the frame was skipped. This process continued for each seedling until at least 15 A and 15 NA stomata were counted.

## Analysis

For each single switch construct, the percentage of seedlings in each of the three categories was plotted in a bar plot with the number of seedlings tested mentioned at the top of the bar. For the full recorders, the percentage of seedlings in each of the expanded seven categories (Supplementary Figs. 5, 7) was plotted but consolidated into the same three categories for display in Figs. 2, 3.

For quantifying the relationship between stomata type and recorder output, the percentage of State 2 and State α stomata of each type was plotted. For comparisons between A and NA stomata, a student's *t* test was performed to evaluate statistical significance.

Python data analysis script which includes statistical tests and plotting functions was run in version 3.9.1 and with the following package dependencies: pandas (version 1.5.3), scipy.stats (version 1.10.0), matplotlib.pyplot (version 3.6.3), matplotlib.colors (version

3.6.3), scikit_posthocs (version 0.21), seaborn (v0.12.0), and numpy (version 1.24.2).

All images taken during seedling characterization were opened and processed using the ImageJ FIJI program (version 1.53c). Each.tif image file contained the images of a seedling's GFP, RFP, and CFP channels. tif files were processed with adjustments to the color lookup table, brightness, and contrast of each channel (GFP: Green, Min 200, Max 2500) (RFP: Red, Min: 300, Max: 3000) (CFP: Blue, Min: 100, Max: 3000).

## Reporting summary
Further information on research design is available in the Nature Portfolio Reporting Summary linked to this article.

## Data availability
Fully annotated sequences of each of the DNA constructs used in this study are available at Benchling [https://benchling.com/cjmaranas/f_/P8Coz4Vw-maranas-et-al-2024-a-history-dependent-integrase-recorder-of-plant-gene-expression-with-single-cell-resolution-/]. Constructs and seed lines used in this study are available by request from J.L.N. (jn7@uw.edu; please expect a response within 3 weeks). DNA constructs can be purchased from Addgene. Arabidopsis seeds are available through the Arabidopsis Biological Resource Center (ABRC). Addgene deposit numbers and ABRC stock numbers are listed in Supplementary Data 2. Source data are provided with this paper and through Figshare [https://doi.org/10.6084/m9.figshare.26824825]. Source data are provided in this paper.

## Code availability
The plotting scripts used in this study have been deposited in Zendo [https://doi.org/10.5281/zenodo.13864568].

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

## Acknowledgements

We thank Janet Solano Sanchez, Ben Downing, and Dr. Alexander Leydon, as well as other members of the Nemhauser, Imaizumi, Di Stilio, Steinbrenner, and Patron groups, for feedback and discussion. We thank Eric Yang for developing and gifting the pPP2AA3 promoter, the Orzáez lab for the PhiC31 sequence, the Brophy lab for the mCherry sequence, and members of the Bonnet lab for sending us the Bxb1 integrase plasmid. We thank Jonah C. Chu for the construction of early versions of the targets and integrase constructs. We thank Keiko Torii for the discussions and advice about stomatal development. This work was supported by grants from the National Institutes of Health (grant no. GM107084, J.L.N.), the National Science Foundation (grant no. IOS-1546873, J.L.N.), and the Howard Hughes Medical Institute Faculty Scholars Program. In addition, support to S.G. was provided by the UK Research and Innovation (UKRI) Biotechnology and Biological Sciences Research Council (BBSRC) via the Earlham Institute Core Capability Grant (grant no. BB/CCG2220/1, S.G.).

## Author contributions

C.M., S.G., and J.L.N. designed the project. C.M., W.G., and S.G. designed the constructs. C.M., W.G., and S.S. generated the constructs. C.M., S.S., and S.V. performed and analyzed the constitutive integrase switching experiments, C.M. performed and analyzed the lateral root and stomata differentiation experiments, C.M., S.G., and J.L.N. wrote the manuscript.

## Competing interests

The authors declare no competing interests.
