## [Peer Review file · Nature Communications]

A history-dependent integrase recorder of plant gene expression with single cell resolution

Corresponding Author: Dr Sarah Guiziou

Version 0:

Reviewer comments:

Reviewer #1

(Remarks to the Author)

DNA based recording systems as synthetic memory switches in cells have become increasingly popular in non—plant systems to overcome challenges based on omic and microscopic techniques. Now they have also entered the plant field providing interesting insights on temporal and spatial gene expression in decoding developmental programs. Therefore this manuscript presents a timely demonstration of the value of this strategy. A prototypic strategy has been developed following the concept of engineered recorder systems in bacterial cells and gaining first insights on prototypes in *Nicotiana*. Work has been extended here into a four state history dependent integrase-based recorder of gene expression in *Arabidopsis*. Construction details, and careful analysis of the different stages is presented. It is indeed shown that synthetic memory circuit switches using tissue specific integrase expression recorder systems enable tracking gene expression in two different developmental programs, ie lateral root initiation and stomata development. Without repeating details important; in my opinion even exciting conclusions can be drawn for both developmental programs. However for the standard reader not fully aware on all the technical details it may significantly help to understand the output between hypothesis and results if at the end of each chapter a careful reviewing assessment will guide the reader in understanding the value of the complex strategy. Otherwise the manuscript is sound deserving publication in NComm.

Reviewer #2

(Remarks to the Author)

In the manuscript entitled 'A history-dependent integrase recorder of plant gene expression with single cell resolution', the authors established a four-state recorder of gene expression history using the integrases PhiC31 and Bxb1, and successfully used the system to test four gene expression histories in the processes of lateral root initiation and stomatal development by using the corresponding promoters to drive the integrases' expression. Overall, the study is novel, with a potential in agricultural application. Here I have a few comments for the study.

1. I like the idea of using integrases and fluorescent reporters to record the gene expression history. However, it seems that in both contexts, a large proportion of samples failed to show 'expected' pattern. When analyzing well-studied promoters, one can propose 'as-expected' pattern, but when studying understudied genes, it will be challenging to interpret the highly variable results. I understand generating homozygous T3 lines is time consuming. However, since the new system presented in the manuscript still needs improvement in stability, it will be a good idea to use clean genetic background to reduce the variability from genetic segregation.
2. When generating the pAHP6 single integrase switch, the authors added an NLS onto PhiC31 to increase its DNA recombination activity. DST was added to the Bxb1 construct in the Bxb1-mediated single integrase switch in the lateral root initiation context. These NLS or DST tags were not always used. Please explain how to decide when to use these tags in the study.
3. This study heavily relies on the promoters to analyze the genes' expression. However, it is a little risky to firmly conclude something solely based on the empirically selected promoters. It may be okay for the well-known promoters such as AHP6, GATA23, SPCH, and MUTE. But when studying new genes, other means of evidence will be needed to verify the function of the selected promoters.

Overall, this manuscript presented a novel system to record gene expression history, and the findings of two different populations of stomata are interesting. I suggest the authors consider my comments before publication.

Reviewer #3

(Remarks to the Author)

The manuscript by Maranas et al represents an important advance in using recombinases as memory switches in plants, this time as a strategy to establish cell lineages. The main novelty is the design of the recorder, which incorporates recognition sites for two integrases, PhiC31 and Bxb1. In this way, the final configuration of the recorder reports the sequential order of activation of the two recombinases. The authors apply this device to the study of the sequential activation of genes involved in the development of lateral roots and the formation of stomata. In both cases, the work shows that using this type of device it is possible to generate stable plants in which the event log agrees with what is expected. However, this expected behaviour is far from being the majority result. A high proportion of the plants obtained have behaviours that the authors label as “underswitched” or “overswitched”, which are far from what was expected. Perhaps one of the most notable contributions of the work is precisely to catalogue these non-as-expected behaviours, which show to what extent current technology is far from being able to generate synthetic devices with robust behaviour like that observed in biological systems. Despite this, the manuscript represents an important advance in the design of this type of recorder. It is clearly written and makes an important impact in the field. Below are a series of suggestions to improve the manuscript as well as issues that the authors should clarify and/or improve.

General comments:

1. In the current state of the design, a good proportion of the lineages obtained present unexpected behaviours. Positional effects are an obvious reason for this. However, the fact that after selecting T1 lines with good behaviour, a high proportion of unexpected behaviours continue to appear in T2 is worrisome. The authors point to the gene dose as a possible cause, and obviously, this can only be elucidated when the gene dose is fixed, so it would be interesting to evaluate the behaviour of at least one generation of plants with the device completely fixed in homozygosity.
2. Along the same lines as the previous comment, the authors should comment on whether, with the current variability, it is possible to use this type of device as a discovery tool, and if so, in what way it would be possible to filter the “overswitched” and “underswitched” so that valuable information can be extracted for the elucidation of gene activation sequences during development. Or if, on the contrary, the current noise level would make its use impossible for systems for which prior information is limited and therefore the improvement of the devices is essential.
3. The authors make some attempts to optimize the recombinase expression cassettes in the single input phase. In the case of PhiC31 they introduce a nuclear localization sequence to increase activity, and in the case of Bxb1 they use an RNA destabilization tag. This indicates that the authors consider, at least in the case of Bxb1, that the basal levels of the recombinase are decisive for the strategy. In that sense, these basal levels are conferred by the promoter of the pair of genes studied. Do the authors think that an optimization of the system is necessary for each pair of genes under study, or does the system only require generic optimization of the sequences of the two recombinases?
4. Besides positional effects and gene dosage, variation can occur due to the intrinsic noise of the device. In that case, one would expect that, in addition to line-to-line and plant-to-plant variation, within-plant variation would occur. Did the authors inspect and compare several parts/organs of the same plant? For instance, in the case of stomata development, did authors inspect several leaves within the same plant? Did all stomata observed in “as expected” plants show “as expected” fluorescence patterns? In other words, did authors observe mixed behaviours (meaning combinations of “as expected”, “over-” “underswitched” fluorescent patterns in stomata within the same plant?

Detail/minor comments:

Line 106 and so on: I suggest using quotation marks to refer to target names, as in the “PhiC31 then Bxb1” target.

Line 115: To facilitate reader understanding, I suggest starting the new paragraph here, as all the following text is devoted to the characterization of the first target.

Lines 187-191. Why did the authors not use selection in this experiment? Did they check afterwards the absence of the integrase TDNA on the unswitched seedlings?

Line 220. Table S1 is very important to follow the course of the story, and the fact that is in the supplementary material makes it difficult to follow. I suggest that authors include a reduced version of Table S1 (containing only the most relevant info and particularly the expected outputs) in the main manuscript body.

Line 244: the sentence “because subsequent generations are likely to have stronger integrase expression” needs explanation: is this due to homozygous gene dosage?

Line 250: Why the authors did not add the DST to Bxb1 in the full lateral root recorder if they found in the individual expression of Bxb1 that it was advantageous to reduce the number of overswitched lines?

Line 258: Fig 2A, B, D. Change labels 1, 2 and 3 to line 1, line 2 and line 3 (we assume numbers refer to lines). Moreover, identifying lines individually with unique names is highly advisable for future exchange of materials with other researchers. The same applies to Figure 3. Also, in the same figure, please indicate clearly that the percentages shown in the figure refer to “as expected” seedlings. Preferable, show percentages for all three categories

Reviewer #4

(Remarks to the Author)

"I co-reviewed this manuscript with one of the reviewers who provided the listed reports. This is part of the Nature Communications initiative to facilitate training in peer review and to provide appropriate recognition for Early Career Researchers who co-review manuscripts."

Version 1:

Reviewer comments:

Reviewer #1

(Remarks to the Author)

The authors have properly addressed the comments of reviewers and improved the manuscript. This is an excellent, innovative piece of work perfectly fitting in the scope of NComm.

My recommendation: publish.

Reviewer #2

(Remarks to the Author)

I appreciate the authors' response to my questions. According to their reply, it seems that the variability of the designed device occurs due to many factors, including the samples' homozygosity, the line-to-line difference, the plant-to-plant difference, the gene dosage variation, the presence of the 'tuning parts', and the intrinsic instability of the designed system (single or dual integrase constructs). The authors also agreed that the presented device needs improvement and has limitations for less studied genes. Although the design of tracking gene expression history using the integrases' activity is novel and interesting, and I believe the manuscript warrants publication somewhere, the current form is not ready for publication in Nature Communications.

Reviewer #3

(Remarks to the Author)

The authors have responded satisfactorily to all previous concerns, and they have implemented the corresponding modifications in text and figures, therefore I can recommend the publication of this interesting manuscript in its present form. Although not affecting the findings and conclusions of the paper, for future studies this reviewer recommends that the authors verify (e.g. by PCR) the presence or absence of the integrase if they opt not to include selection on the plates (or work with homozygous T3 population for characterization). As mentioned in their response, in a previous study (Supplementary Figure 8) it is shown that at least a small proportion of resistant seedlings still exhibit the unswitched phenotype. Therefore, removal of the unswitched seedlings under the assumption that they will be azygous is acceptable, but not best practice.

Reviewer #4

(Remarks to the Author)

We thank all the reviewers for their time and effort in reading and offering advice and comments on our manuscript. We addressed each comment and made edits in the manuscript accordingly. We believe this revised manuscript is improved on multiple fronts thanks to the thoughtful reviewer comments. Below please find our point by point response to reviewers.

Response to Reviewer #1:

1. "However for the standard reader not fully aware on all the technical details it may significantly help to understand the output between hypothesis and results if at the end of each chapter a careful reviewing assessment will guide the reader in understanding the value of the complex strategy."

- We thank the reviewer for supporting the publication of our manuscript and for advice on improving the clarity of the results in the text.

We added or edited the following statements at the end of each chapter to clarify the results for the readers:

(line 139) "We designed and implemented history-dependent integrase targets in *Arabidopsis* with identifiable fluorescent outputs for the different DNA states. "

(line 274) "Overall, we implemented a recorder tracking the expression patterns of *AHP6* and *GATA23* during lateral root initiation and consistently generated T2 seedlings with the expected switch pattern."

(line 387) "Despite the observed variability, we consistently generated seedlings in the 'as expected' category for the stomata recorder and the *SPCH* and *MUTE* single integrase switches."

Additionally, from what we understood, the reviewer believes the section "The history dependent recorder can identify cells that have undergone an alternate developmental path" would benefit from additional contextualization of the results with respect to our hypothesis. To address this, we have added a sentence at the end of this section to contextualize the results of the stomata counting experiment with our hypothesis:

(line 476) "Overall, we showed that the state of the recorder in a given stoma correlated strongly with its developmental history and stomatal type. The successful encoding of differences in the relative timing of *SPCH* and *MUTE* expression during alternative developmental trajectories demonstrates the potential utility of this strategy in connecting patterns of gene expression to differentiation in many other contexts."

Reviewer #2:

Thank you to the reviewer for their positive comments and support for the publication of our manuscript. We appreciated the constructive comments and have responded to each with appropriate explanations and/or edits to the manuscript.

1. "I like the idea of using integrases and fluorescent reporters to record the gene expression

history. However, it seems that in both contexts, a large proportion of samples failed to show 'expected' pattern. When analyzing well-studied promoters, one can propose 'as-expected' pattern, but when studying understudied genes, it will be challenging to interpret the highly variable results. I understand generating homozygous T3 lines is time consuming. However, since the new system presented in the manuscript still needs improvement in stability, it will be a good idea to use clean genetic background to reduce the variability from genetic segregation."

- We thank the reviewer for bringing up this point and we agree that our recorder would benefit from improvements in stability. It is worth noting that in plants with overswitching (the majority of the not "as expected" plants), the integrase activity does still reflect gene expression. The 'overswitching' phenotype arises because the integrase is so efficient that even basal levels of gene expression can be enough to trigger the switch. We added the following parts to the text to clarify this distinction:

(line 161) "To note, the integrase recorder captures gene expression in a digital manner, where each gene is either expressed (1) or not expressed (0). This is in contrast to biological systems where gene expression is analog. Gene expression levels are continuous with a cell-type-specific distribution of expression around a mean. As a result, expression levels that might be considered background or basal can be sufficient for integrase activity. In practice in the integrase recorder, the digital state 0 corresponds to low gene expression and the digital state 1 corresponds to expression above a specific threshold. This threshold is dependent on the level of integrase activity, and tuning parts can be used to adjust the switch threshold for the expression level of the gene of interest."

(line 199) "To note, in our system overswitching corresponds to switching happening at basal expression levels, meaning the integrase switch threshold is too low. Underswitching corresponds to an integrase switch threshold that is too high resulting in fewer cells than expected undergoing the switch."

To the reviewer's point regarding the benefit of characterization in a clean genetic background, we performed characterization of six T3 lines coming from our best performing stomatal recorder T2 line (T2P1). We characterized the switch patterns of T3 seedlings within each line and in tandem, grew the T3 seedlings on hygromycin selection to determine if the line was a homozygous line (100% hygromycin resistance) or not (75% hygromycin resistance). Of those six lines, three showed switching so we focused our analysis on these lines. The other three lines had either the target or integrase construct missing or silenced. We added a supplemental figure (Figure S8) with the results. Of the three lines, one (T3P1.4) is homozygous for the integrase construct and this line showed a higher rate of overswitching compared to the other two lines which are comprised of individuals with varied integrase copy number. This finding strengthens our assertion that integrase copy number contributes to variability within T2 lines so, in theory, a more carefully tuned homozygous line should switch optimally and will have lower variability between seedlings. There still exists some variation in this line, perhaps indicating that some of the variation captured by the recorder is biological, caused by natural variations in promoter expression level. Additionally, we found that the

switch pattern for the two non-homozygous lines (T3P1.1 and T3P1.3) mimicked that of the parental T2 line.

2. “When generating the pAHP6 single integrase switch, the authors added an NLS onto PhiC31 to increase its DNA recombination activity. DST was added to the Bxb1 construct in the Bxb1-mediated single integrase switch in the lateral root initiation context. These NLS or DST tags were not always used. Please explain how to decide when to use these tags in the study.”

- We thank the reviewer for this feedback, as the choice of using tuning parts is crucial to achieving a specific and robust integrase switch. The approach we took starts with generating the integrase construct without any tuning parts. Then based on the switch result with this construct, we added a tuning part if needed (NLS if not switching enough and DST if switching non-specifically). To make this clear in the text, we added the following:

(line 171)“The approach we took for deciding which tuning parts to use starts with generating the integrase construct without any tuning parts. Then, based on the switch result with this construct, we added a tuning part if needed, either a nuclear localization signal tag (NLS, shown to increase integrase switching efficiency) if underswitched or an RNA destabilization tag from SMALL AUXIN UP-REGULATED RNA genes (DST) if overswitched.”

Another consideration is that we expected that the second integrase (Bxb1) in the dual integrase construct will be less expressed due to transcription induced supercoiling of the upstream PhiC31 gene⁴⁶. Indeed, we observed this to be the case, particularly for the pGATA23::Bxb1 integrase switch whose efficiency was reduced in the dual integrase construct as compared to the single pGATA23::Bxb1 construct. Using this dual construct, we consistently observed seedlings with ‘as expected’ Bxb1 switching, whereas in the single pGATA23::Bxb1 construct with no tuning parts we observed only overswitched seedlings. The reviewer hits on a good point that a more defined guide for when to use each tuning part would be quite beneficial for these applications. This would be aided by characterization of each integrase’s efficiency and the effect of each tuning part, such that this information, in combination with the expression level of the promoter of interest, can inform an ideal integrase construct to obtain switching at the expression level of interest. Such an analysis is beyond the scope of the current manuscript, and we added a brief explanation of this consideration in the discussion:

(line 543) “In the future, information from a thorough characterization of integrase efficiency combined with knowledge of gene expression strength could inform the choice of tuning parts, expediting the process for generating robust and specific integrase switches.”

3. “This study heavily relies on the promoters to analyze the genes’ expression. However, it is a little risky to firmly conclude something solely based on the empirically selected promoters. It may be okay for the well-known promoters such as AHP6, GATA23, SPCH, and MUTE. But when studying new genes, other means of evidence will be needed to verify the function of the selected promoters.”

- We agree with the reviewer's point that this approach requires preliminary knowledge of the promoter expression which can be done using a simple transcriptional reporter or using transcriptomic data. This information, combined with detailed characterization of the integrase switch in multiple lines and seedlings within lines should allow evaluation of cell-to-cell variation in expression even for less characterized genes.

We also want to emphasize the unique advantages of our system for use with less characterized promoters, such as the sustained fluorescent expression mediated by our recorder in contrast with a traditional reporter and amplification of signal allowing characterization of lowly-expressed genes. Our system also allows determination of cell lineages. In addition, less well studied genes can be combined in our recorder with well studied genes to provide information about their relative expression dynamics (e.g., to test the predictions of a pseudotime trajectory from scRNA-seq). We added the following section into the discussion to address these points and provide guidance for those interested in adapting our approach for less well characterized processes.

(line 525)“While we applied our recorder design to well characterized differentiation processes, it can be adapted to processes wherein the patterns of gene expression are not well understood. This would necessitate additional gene expression validation using a transcriptional reporter or spatial transcriptomics data. With this extra validation step, our recorder offers unique advantages for less well studied genes including: sustained fluorescent expression to enable recording even when the exact timing of expression is unknown; high sensitivity to enable recording even at low levels of expression; the ability to discern variation from cell to cell; and the tracing of the cell lineage which had expressed the gene of interest.”

Reviewer #3 (Remarks to the Author):

- We want to thank the reviewer 3 for their interest towards our paper. We do agree that the scientific community still has some work to do to make this technology robust. We do believe as the reviewer highlighted that this work is an important advance towards robust integrase circuits to track gene-expression at a single cell resolution. Additionally, as highlighted by the reviewer, we do agree that the characterisation of numerous lines and seedlings allow precise characterization of those systems and is an important step towards designing integrase circuits with robust switching output.

General comments:

1. “In the current state of the design, a good proportion of the lineages obtained present unexpected behaviours. Positional effects are an obvious reason for this. However, the fact that after selecting T1 lines with good behaviour, a high proportion of unexpected behaviours continue to appear in T2 is worrisome. The authors point to the gene dose as a possible cause, and obviously, this can only be elucidated when the gene dose is fixed, so it would be interesting to evaluate the behaviour of at least one generation of plants with the device completely fixed in homozygosity.”

- We thank the reviewer for this feedback and agree that characterization in stable lines would be beneficial. As highlighted additionally by reviewer #2, we characterized T3 seedlings and analyzed their zygosity (Fig. S8). Of the three T3 lines we analyzed which showed switching, one was homozygous and the other two were non-homozygous. The homozygous line showed a much higher rate of overswitching than the parental T2 line, showing the effect of copy number on integrase switch efficiency and implying that the variation in integrase copy number in the non-homozygous T2 and T3 lines contributes to variability. We expect that a more optimally tuned homozygous line would show optimized switching and lower variation between seedlings. Please also see our response to Reviewer #2, point #1 as they had a similar comment.

2. “Along the same lines as the previous comment, the authors should comment on whether, with the current variability, it is possible to use this type of device as a discovery tool, and if so, in what way it would be possible to filter the “overswitched” and “underswitched” so that valuable information can be extracted for the elucidation of gene activation sequences during development. Or if, on the contrary, the current noise level would make its use impossible for systems for which prior information is limited and therefore the improvement of the devices is essential.”

- We feel strongly that this system has applications as a discovery tool, albeit with some extra validation of promoter expression patterns using a transcriptional reporter or transcriptomic data, so we appreciate the opportunity to state that more clearly. This technology can be used in its current state as a discovery tool for cell lineages or to reveal relative expression patterns when the gene is tracked using the recorder along with a second well characterized gene. But we do agree that improvement in switch robustness is important to use only this device as a discovery tool, so efforts to generate homozygous, properly tuned lines will be particularly important for less studied genes. Please refer to our response to Reviewer #2, point #3 for more discussion of this point.

3. “The authors make some attempts to optimize the recombinase expression cassettes in the single input phase. In the case of PhiC31 they introduce a nuclear localization sequence to increase activity, and in the case of Bxb1 they use an RNA destabilization tag. This indicates that the authors consider, at least in the case of Bxb1, that the basal levels of the recombinase are decisive for the strategy. In that sense, these basal levels are conferred by the promoter of the pair of genes studied. Do the authors think that an optimization of the system is necessary for each pair of genes under study, or does the system only require generic optimization of the sequences of the two recombinases?”

- Choice of tuning parts is an important consideration when designing an integrase switch and the reviewer is correct that the strength of the promoters for the genes studied (along with the integrase efficiency) defines the need, or lack thereof, for the addition of tuning parts. At this point, each set of genes of interest does require its own optimization. However, future efforts to define the integrase efficiency and the effect of each tuning part could be combined with existing information on promoter expression strength to generate an informed recommendation for tuning parts given a set of genes of interest.

This way, the optimization process can be streamlined and allow faster generation of properly behaving recorders. We have added this section in the discussion to address this point:

(line 543)“In the future, information from a thorough characterization of integrase efficiency combined with knowledge of gene expression strength could inform the choice of tuning parts, expediting the process for generating robust and specific integrase switches.”

4. “Besides positional effects and gene dosage, variation can occur due to the intrinsic noise of the device. In that case, one would expect that, in addition to line-to-line and plant-to-plant variation, within-plant variation would occur. Did the authors inspect and compare several parts/organs of the same plant? For instance, in the case of stomata development, did authors inspect several leaves within the same plant? Did all stomata observed in “as expected” plants show “as expected” fluorescence patterns? In other words, did authors observe mixed behaviours (meaning combinations of “as expected”, “over-” “underswitched” fluorescent patterns in stomata within the same plant?”

- We thank the reviewer for the idea of considering intra-plant variation in our characterization. To answer the question, no we did not observe significant differences in recorder output within the same plant. In the case of the lateral root recorder, within one seedling, every lateral root of a similar developmental stage showed the same recorder output whether it be underswitched, overswitched, or “as expected”. In the stomata recorder, all the stomata were either in State 2 or State α and that variation is explained by disparate developmental paths as outlined in Figure 4. We did not characterize multiple leaves from the same seedling, but we did characterize across all locations on one leaf and observed negligible variation. This intra-plant consistency also extends to all of the single integrase switches. We added short sections to the text of the results outlining the observed intra-plant variation, with respect to the lateral root:

(line 270)“We observed negligible intra-plant variation in the fluorescent output of the *AHP6* single integrase switch, the *GATA23* single integrase switch, and the full lateral root recorder. Lateral roots of similar developmental stages on the same seedling showed the same fluorescent output, indicating low levels of intrinsic noise in the integrase switches.”

And with respect to stomata:

(line 359)“For both of the single integrase switches, we observed negligible variation in the switch pattern across locations on the leaf, with leaf epidermal cells throughout showing a consistent switch output, whether it was underswitched, overswitched, or “as expected”.”

Detail/minor comments:

Line 106 and so on: I suggest using quotation marks to refer to target names, as in the “PhiC31 then Bxb1” target.

- done, thanks for the suggestion.

Line 115: To facilitate reader understanding, I suggest starting the new paragraph here, as all the following text is devoted to the characterization of the first target.

- done, thanks for the suggestion.

Lines 187-191. Why did the authors not use selection in this experiment? Did they check afterwards the absence of the integrase TDNA on the unswitched seedlings?

- We did not use selection because it is known that antibiotics cause stress to the plant which could impact plant development and underlying gene expression dynamics. In our previous publication²², Supplementary Figure 8 shows characterization of integrase switch lines with subsequent evaluation of hygromycin resistance (to determine the presence of the integrase construct). The resistant seedlings showed a much lower rate of non-switching seedlings compared to the total population, indicating that a significant proportion of the unswitched population is missing the integrase construct. We see no reason this effect would be different for this manuscript so we did not repeat this characterization.

“Line 220. Table S1 is very important to follow the course of the story, and the fact that is in the supplementary material makes it difficult to follow. I suggest that authors include a reduced version of Table S1 (containing only the most relevant info and particularly the expected outputs) in the main manuscript body. “

- We appreciate the input on the helpfulness of the table and have taken the reviewer’s suggestion to move it to the main text by creating a simplified version (Table 1) and leaving the more detailed Table S1 in the supplement.

Table 1: Overview of integrase lines and characterization categories

	As expected	Underswitched	Overswitched
pAHP6 single switch	RFP in xylem and lateral roots BFP otherwise	RFP in less cells than xylem and LR	RFP in more cells than xylem and LR
pGATA23 single switch	RFP in LR BFP otherwise	RFP in less cells than LR	RFP in more cells than LR
Lateral root history-dependent tracker	GFP in LR, RFP in xylem, BFP otherwise	GFP in less cells than LR and/or RFP in less cells than xylem	GFP in more cells than LR and/or RFP in more cells than xylem and LR
pSPCH single switch	RFP in guard cells and surrounding epidermal cells, BFP otherwise	RFP in less cells than guard cells and surrounding epidermal cells	RFP in more cells than guard cells and surrounding epidermal cells

pMUTE single switch	RFP in guard cells BFP otherwise	RFP in less cells than guard cells	RFP in more cells than guard cells
Stomatal history-dependent tracker	GFP in guard cells RFP surrounding epidermal cells, BFP otherwise	GFP in less cells than guard cells and/or RFP in less cells than surrounding epidermal cells	GFP in more cells than guard cells and/or RFP in more cells than surrounding epidermal cells

Line 244: the sentence “because subsequent generations are likely to have stronger integrase expression” needs explanation: is this due to homozygous gene dosage?

- Yes, due to the gene dosage. We adjusted the phrasing to clarify this:
(line 260)“because subsequent generations are likely to have stronger integrase expression due to higher copy number in homozygous individuals”

Line 250: Why the authors did not add the DST to Bxb1 in the full lateral root recorder if they found in the individual expression of Bxb1 that it was advantageous to reduce the number of overswitched lines?

- Expression strength varies between the single integrase constructs and the dual integrase constructs, particularly in the case of the downstream Bxb1 gene whose expression is reduced compared to the single Bxb1 constructs. This is why we were able to achieve *GATA23*-specific switching with no tuning parts in the dual integrase construct but not with the single Bxb1 integrase construct (see Table S2). However, the reviewer is correct that addition of a DST to the dual integrase lateral root recorder construct would likely further reduce the Bxb1 overswitching but would likely increase the proportion of Bxb1 underswitching seedlings. Please see our response to Reviewer #2, point #2 for further discussion on the choice of tuning parts.

Line 258: Fig 2A, B, D. Change labels 1, 2 and 3 to line 1, line 2 and line 3 (we assume numbers refer to lines). Moreover, identifying lines individually with unique names is highly advisable for future exchange of materials with other researchers. The same applies to Figure 3. Also, in the same figure, please indicate clearly that the percentages shown in the figure refer to “as expected” seedlings. Preferable, show percentages for all three categories

- We have made the suggested edits to Figures 2 and 3 to include “line” in the labels and to add the percentages for each switch category. The unique name of each line is specified in the legend for each of the figures.

Reviewer #4 (Remarks to the Author):

"I co-reviewed this manuscript with one of the reviewers who provided the listed reports. This is part of the Nature Communications initiative to facilitate training in peer review and to provide appropriate recognition for Early Career Researchers who co-review manuscripts."

- Thanks for giving this opportunity to an Early Career Researcher.

We thank all the reviewers for their time and effort in reading and offering advice and comments on our manuscript. We addressed each comment and made edits in the manuscript accordingly. Below please find our point by point response to reviewers.

Response to Reviewer #1:

"The authors have properly addressed the comments of reviewers and improved the manuscript. This is an excellent, innovative piece of work perfectly fitting in the scope of NComm. My recommendation: publish."

We thank Reviewer #1 for supporting the publication of our work.

Response to Reviewer #2:

"I appreciate the authors' response to my questions. According to their reply, it seems that the variability of the designed device occurs due to many factors, including the samples' homozygosity, the line-to-line difference, the plant-to-plant difference, the gene dosage variation, the presence of the 'tuning parts', and the intrinsic instability of the designed system (single or dual integrase constructs). The authors also agreed that the presented device needs improvement and has limitations for less studied genes. Although the design of tracking gene expression history using the integrases' activity is novel and interesting, and I believe the manuscript warrants publication somewhere, the current form is not ready for publication in Nature Communications."

We thank Reviewer #2 for their comments. We agree that the line-to-line and plant-to-plant variability will have to be improved for applications using less studied genes. We edited the following sentence to make this clear in the manuscript: "This would necessitate additional optimization of the integrase switch specificity to reduce plant-to-plant and cell-to-cell variability combined with gene expression validation using a transcriptional reporter or spatial transcriptomics data." Line 451. However, as shown in Figure 4, this history-dependent recorder allows the recording of the variability of order of gene expression in the context of two different stomata developmental pathways. This is one example of its current potential applications. This technology without further optimisation allows to marking of specific cell types and cell lineages which would provide a wide range of applications from understanding to engineering gene regulatory networks in plants.

Response to Reviewer #3:

"The authors have responded satisfactorily to all previous concerns, and they have implemented the corresponding modifications in text and figures, therefore I can recommend the publication of this interesting manuscript in its present form.

Although not affecting the findings and conclusions of the paper, for future studies this reviewer recommends that the authors verify (e.g. by PCR) the presence or absence of the integrase if they opt not to include selection on the plates (or work with homozygous T3 population for characterization). As mentioned in their response, in a previous study (Supplementary Figure 8) it is shown that at least a small proportion of resistant seedlings still exhibit the unswitched phenotype. Therefore, removal of the unswitched seedlings under the assumption that they will be azygous is acceptable, but not best practice."

We thank Reviewer #3 for their suggestions, and we will make sure we present the results of the homozygous T3 population in future work. We added the following statement in the discussion to specify this: "For future applications, characterization and optimization of T3 homozygous lines will be required" line 473. One of the objectives of showing the characterisation of T2 seedlings is also to inform how many lines and seedlings need to be tested to obtain the desired phenotypes. In many papers, only the data of the best homozygous line is presented, we think that showing the characterisation of a wide range of lines is more informative.

We thank Reviewer #3 for supporting the publication of our work.

Response to Reviewer #4:

"I co-reviewed this manuscript with one of the reviewers who provided the listed reports. This is part of the Nature Communications initiative to facilitate training in peer review and to provide appropriate recognition for Early Career Researchers who co-review manuscripts."

Thanks for giving this opportunity to an Early Career Researcher.